OPINION/HYPOTHESIS

# Ancient Metagenomic Studies: Considerations for the Wider Scientific Community

Clio Der Sarkissian,[a] Irina M. Velsko,[b] Anna K. Fotakis,[c] Åshild J. Vågene,[d] Alexander Hübner,[b,e] James A. Fellows Yates[b,f,g]

[a]Centre for Anthropobiology and Genomics of Toulouse, UMR5288, CNRS, University Toulouse 3 Paul Sabatier, Toulouse, France
[b]Department of Archaeogenetics, Max Planck Institute for Evolutionary Anthropology, Leipzig, Germany
[c]Center for Evolutionary Hologenomics, GLOBE Institute, Faculty of Health and Medical Sciences, University of Copenhagen, Copenhagen, Denmark
[d]Section for Evolutionary Genomics, GLOBE Institute, Faculty of Health and Medical Sciences, University of Copenhagen, Copenhagen, Denmark
[e]Faculty of Biological Sciences, Friedrich-Schiller University Jena, Jena, Germany
[f]Institut für Vor- und Frühgeschichtliche Archäologie und Provinzialrömische Archäologie, Ludwig Maximilian University Munich, Munich, Germany
[g]Department of Paleobiotechnology, Leibniz Institute for Natural Product Research and Infection Biology Hans Knöll Institute, Jena, Germany

Clio Der Sarkissian, Irina M. Velsko, Anna K. Fotakis, Åshild J. Vågene, Alexander Hübner, and James A. Fellows Yates wrote the manuscript together. All authors contributed equally, and author order was selected by lottery.

**ABSTRACT**   Like modern metagenomics, ancient metagenomics is a highly data-rich discipline, with the added challenge that the DNA of interest is degraded and, depending on the sample type, in low abundance. This requires the application of specialized measures during molecular experiments and computational analyses. Furthermore, researchers often work with finite sample sizes, which impedes optimal experimental design and control of confounding factors, and with ethically sensitive samples necessitating the consideration of additional guidelines. In September 2020, early career researchers in the field of ancient metagenomics met (Standards, Precautions & Advances in Ancient Metagenomics 2 [SPAAM2] community meeting) to discuss the state of the field and how to address current challenges. Here, in an effort to bridge the gap between ancient and modern metagenomics, we highlight and reflect upon some common misconceptions, provide a brief overview of the challenges in our field, and point toward useful resources for potential reviewers and newcomers to the field.

**KEYWORDS**   ancient metagenomics, ancient DNA, archaeogenomics, paleogenetics, archaeology, microbiome, metagenomics, microbe, authentication, ethics, cultural heritage, peer review

## BACKGROUND

Ancient metagenomics refers to the analysis of the complex DNA content recovered from degraded, nonliving, biological material (e.g., bones, teeth, dental plaque, paleofeces, sediments), primarily via shotgun high-throughput sequencing. Research often focuses on ancient microbes (1, 2) but increasingly also on the simultaneous analysis of numerous (extinct) macroorganisms (see, e.g., references 3 to 5). Research questions in the field are often highly interdisciplinary, spanning the humanities and social and natural sciences, adding new perspectives to our understanding of the past, e.g., characterizing causative candidates of historical epidemics (6), identifying oral microbes in Neanderthals (7), reconstructing postglacial animal and plant successions in North America (8), and integrating detailed social and cultural contexts from archaeology and history (9, 10). Analyzing ancient metagenomes can provide a wide range of exciting and complementary sources of information for modern metagenomic studies, such as millennial-scale insights into the dynamics of metagenomes and the evolution of microbial species through deep-time calibration points for phylogenetic analyses (11–13). However, ancient metagenomics faces particular logistical and molecular challenges in addition to those commonly met when analyzing modern samples.

Address correspondence to James A. Fellows Yates, james_fellows_yates@eva.mpg.de.

The authors declare no conflict of interest.

mSystems®

**TABLE 1** Common terms used in ancient metagenomics

| Term | Description | Relevant literature |
|---|---|---|
| SPAAM | Standards, Precautions, and Advances in Ancient Metagenomics, a community of researchers in ancient metagenomics who run regular discussion meetings and community projects. | spaam-community.github.io |
| Ancient | An organism or tissue is often considered to qualify as ancient when it is more than 100 years old. This cutoff, however, is arbitrary. | |
| Dental calculus | Mineralized dental plaque, also known as dental tartar, contains the petrified remains of an oral biofilm of microorganisms and other micro- and macroremains derived from the oral cavity. | 7, 47, 55, 56 |
| Paleofeces | Ancient human or animal feces that remain in an organic or partially organic state. Completely mineralized fecal remains are termed coprolites (however, note that the terms are often used interchangeably in archaeology). | 57 |
| Historic specimen | Any specimen that is no longer living and can no longer undergo molecular repair mechanisms. Sometimes used to distinguish natural history museum specimen collections aged less than 200 years old from mostly mineralized tissues found in archaeological excavations older than 200 years. | |
| Bio-cultural heritage | Biological specimens, tissues, or secondary substrates (e.g., dental calculus and paleofeces) or the habitats that are derived from (human) culture that are of cultural significance to a society or individual. | 58 |
| Necrobiome | The (micro)organisms that decompose dead organic materials, mostly referring to dead bodies. | 59–61 |
| Postdepositional | Changes or contamination affecting the specimen after deposition in the ground or environmental context not used during life (e.g., burial). | |
| Degradation | The process of biomolecules being broken up and damaged through a variety of chemical and mechanical processes. | 62 |
| Molecular preservation | State of the chemical preservation of the biomolecules, used to evaluate the feasibility of cost-effective hybridization capture, sequencing, and analysis. | |
| Destructive sampling | Sampling that removes part of the sample or specimen such that the removed/collected part cannot be returned after analysis. | |
| Ancient DNA | DNA from deceased organisms that has undergone some form of postmortem degradation processes (e.g., fragmentation, deamination damage). Often referred to as aDNA. | 63, 64 |
| Fragmentation | Breakage of DNA backbones that normally occurs at sites of base depurination and often results in single-stranded overhangs. Over time, aDNA reads become very short, typically 30–70 base pairs (bp). | 65, 66 |
| Misincorporation | Higher frequency of cytosine deamination-derived uracils (C-to-U) that are read as T by nonproofreading polymerases at DNA termini, caused by hydrolysis. This occurs on exposed single-stranded overhangs of fragmented DNA. | 65, 66 |
| Damage pattern | Can refer to either nucleotide misincorporation or fragment length distributions or both. | 67, 68 |
| Contamination | Ancient and modern DNA not deriving from the original organism or sample of interest (e.g., from the burial environment, museum collection, laboratory environment, curators, researchers). | 41, 69 |
| Endogenous DNA | DNA originally derived from the sample or tissue type that does not come from contaminating sources. | |
| Authentication | Determining whether a given set of DNA molecules is truly ancient and belonging to, or originating from, the sample in question. Normally based on characteristic damage patterns and endogenous content. | |
| Negative controls | Controls that do not contain samples for analysis (typically water) to act as indicators for the laboratory performance of reagents, or protocols and cross-contamination from the laboratory or samples in the same batch. | 41 |
| Radiocarbon dating | A means of annual age determination by measuring the decay of radioactive carbon, which occurs at a consistent rate. Also known as C-14 or carbon dating, with dates typically reported as "before present," where "present" is usually defined as approximately 1950. | 70 |
| FAIR principles | A set of guidelines that focus on sharing data and metadata in a way that is machine readable, easily findable, and reusable to promote reproducibility. | 27 |

These challenges stem from the facts that ancient samples are often rare and precious and that their molecular content is often highly degraded (see Table 1 for definitions of common terms used in the field); the latter makes ancient DNA (aDNA) sensitive to contamination by modern DNA from natural, storage, and experimental environments, including from handling (a selection of review papers can be found in Table 2). To better characterize and mitigate challenges in the field through collaboration, ancient metagenomic researchers organized a workshop for Standards, Precautions, and Advances in Ancient Metagenomics (SPAAM) in 2016 (1). This served as an inspiration to formally establish the SPAAM community in 2020 (https://spaam-workshop.github.io/).

**TABLE 2** Nonexhaustive list of suggested reviews, laboratory comparisons, dedicated analysis tools, and benchmarking studies relevant for ancient metagenomics

| Publication type | Description | Reference |
|---|---|---|
| General review | General but comprehensive introduction to microbial archaeology, including ancient metagenomics and pathogen reconstruction and their challenges. Output of the first SPAAM meeting in 2016. | 1 |
| | Descriptions of the challenges related to contamination in next-generation sequencing data sets of low-biomass samples, including ancient DNA. | 71 |
| | Review of approaches developed for estimating the level of contamination in ancient human DNA data sets, covering aspects also relevant for ancient metagenomics. | 72 |
| | State-of-the-art summary of ancient pathogen research and what can be learned from such genomes. | 73 |
| Laboratory protocol | Comparison of decontamination and aDNA extraction protocols for ancient dental calculus. | 47 |
| | Comparison of decontamination protocols for ancient mammalian dental calculus. | 74 |
| | Comparison of decontamination protocols for ancient human dental calculus. | 75 |
| | Comparison of different protocols for simultaneous extraction of different ancient biomolecules from dental calculus. | 76 |
| | Comparison of DNA extraction methods for paleofeces, including commonly used modern DNA extraction kits. | 39 |
| | Comparison of microbial genome enrichment techniques for ancient pathogens. | 77 |
| | Development of techniques for aDNA retrieval from cave sediments and mammalian DNA capture techniques. | 3 |
| | Extraction method for retrieval of eukaryotic aDNA from marine sediment. | 78 |
| | Capture protocol for enrichment of marine eukaryotic aDNA. | 79 |
| Computational tool | gargammel: synthetic ancient DNA data set generation with a damage and contamination modeling tool. | 54 |
| | MALT: includes a description of the ultrafast BLAST-like metagenomic aligner MALT, which includes adaptations to account for ancient DNA damage. Often used for taxonomic profiling or pathogen detection. | 6 |
| | PIA: a taxonomic read binner used to identify the likely host source of typically sedimentary aDNA (sedaDNA) reads, with a focus on accounting for extinct taxa not present in modern databases. | 80 |
| | SNP_Evaluation: a tool allowing for evaluating and visualizing confidence in variant calling of low-coverage pathogen genomes. | 81 |
| | MEx-IPA: an interactive viewer of output from the HOPS metagenomic authentication pipeline. | 82 |
| | cuperdec: an R package for the estimation and visualization of the endogenous taxonomic content of ancient microbiomes. | 82 |
| | PyDamage: a tool for separating ancient and modern contigs from *de novo* assembly of ancient DNA data. | 43 |
| Pipeline | Holi: a pipeline for taxonomic profiling of ancient metagenomic reads based on competitive mapping to large databases. Often used to profile ancient environmental samples (e.g., sedaDNA) for reads assigned to animal or plant taxa. | 8 |
| | metaBIT: the first pipeline with configurations for high-throughput ancient metagenomic screening with MetaPhlAn and a range of taxonomic profile comparison analyses. | 83 |
| | coproID: a pipeline for the prediction of the host organism of ancient fecal material, including taxonomic profiling of the endogenous content of both host and microbial DNAs. | 84 |
| | HOPS: a pipeline integrating MALT alignment with postalignment ancient DNA characteristic authentication. Includes damage profiling, fragment length, and visualization of possible contamination. | 85 |
| | nf-core/eager: a general ancient DNA genomics pipeline with taxonomic profiling for pathogen screening (and microbiome reconstruction) and authentication components, as well as steps allowing for analysis of microbial genomes. | 86 |
| Benchmarking | Comparison of different metagenomic taxonomic classifiers applied to ancient DNA data, with description of effects of aDNA damage and short fragment lengths. | 87 |
| | Comparison of the effects of nucleotide-to-nucleotide vs nucleotide-to-protein taxonomic classification for short fragment lengths. | 88 |

The SPAAM2 meeting in September 2020 gathered 63 early career researchers representing 36 institutions from 15 countries with highly diverse and interdisciplinary backgrounds (14). During discussions, it became evident that this young field would greatly benefit from building more bridges and improving communication with metagenomic or microbiology researchers working with modern samples. This was particularly brought to light in reviews of grant applications and publications. In this opinion/hypothesis, we highlight a few common misconceptions and point toward useful resources and field basics aimed at potential reviewers, as well as newcomers to the field. Additionally, we reflect on current research in our field and how we can improve on existing challenges to maximize the potential that ancient metagenomics offers.

## SAMPLING STRATEGIES IN AN ANCIENT METAGENOMIC STUDY

Small sample sizes are a consistent feature of ancient metagenomic studies, meaning that there is often little opportunity for large-scale sampling or elaborate sampling design,

as might be expected by reviewers with a background in modern metagenomics. There are several factors at play, including (i) variability in what was recovered from archaeological excavations and saved from collaborations between curators, archaeologists, and other legal representatives who safeguard the material and (ii) what has sufficient molecular preservation for analysis. These are limiting factors that are, largely, out of the control of researchers working on ancient or historical specimens. In particular, the preservation of the physical samples as well as the molecular preservation are strongly affected by the conditions of both the burial and storage environments. Therefore, deeper sequencing can be financially prohibitive, and increasing sample sizes retroactively from the same archaeological or museum collection or resampling previously accessed samples for replicates is not trivial, if even possible. This is partly due to sample accessibility but also due to "destructive sampling"; samples are often small amounts of material that cannot be regenerated like a microbial culture, nor can they be replaced. Due to the rarity and often small quantity of ancient remains, efforts are in fact made to avoid unnecessary, redundant subsampling by restricting access to these materials (15).

While ancient metagenomic researchers generally aim for high-quality sequencing data sets, reviewers should consider whether results and conclusions are supported by the data as they stand within this context. As more ancient metagenomic papers are published, researchers will be able to increase study sample sizes by incorporating previously published data sets and thus mitigate the effect of small sample size. One such example is the unexpected discovery of early forms of *Yersinia pestis* in only 7 out of 101 Bronze Age individuals, with the disease being argued as "endemic" despite geographic sparsity (11). However, this has since been repeatedly verified and expanded, with an increasing number of genomes recovered from the same and new regions (16–20).

Power analyses are likely to indicate that ancient metagenomic studies are always underpowered, but when performed prior to study design, they may clarify what questions can be answered with the data set at hand and guide the study's focus. Further, involving statisticians during study design is encouraged to determine the best methods to test the validity of their findings for robust verification. The questions being asked of ancient data sets need to be tailored for limited sample sizes, and interpreters of results should not make sweeping or broad claims that cannot be accurately extrapolated beyond the current study. As an example, Velsko et al. (21) collected extensive metadata regarding calculus deposition patterns to perform association tests with the calculus taxonomic profiles. However, it became clear that there were insufficient sample numbers in each category to draw statistically supported conclusions about such associations, and instead, the authors then refocused the paper to address other questions. Finally, to mitigate the bias of small sample sizes, the use of appropriate sample randomization for laboratory processing to avoid batch effects, and the sequencing of extraction and library processing negative control experiments need to be performed.

## ETHICAL CONSIDERATIONS AND CULTURAL SIGNIFICANCE

Ancient metagenomicists are faced with specific limitations and requirements for responsible and ethical research conduct when applying (partially) destructive analyses to archaeological materials. These materials are indeed scarce, irreplaceable, and constituent of our bio-cultural heritage, requiring their historical, cultural, social, and political background to be considered, especially those derived from extremely sensitive contexts, e.g., colonialism, racism, exploitation, or stigmatization. Ethical considerations can also apply to environmental samples, such as sediments, in particular those linked with past or present indigenous communities (22).

Currently, there is no internationally agreed-upon framework that specifically oversees genetic work on ancient specimens, and the responsibility of ensuring ethical ancient metagenomic research admittedly falls upon us and collection/material curators. Agreements such as the Convention on Biological Diversity (https://www.cbd.int/) and the subsequent Nagoya protocol (89) exist for genetic and biological material; however, these remain ambiguous with regard to ancient material. There is currently no precedent for intellectual property rights

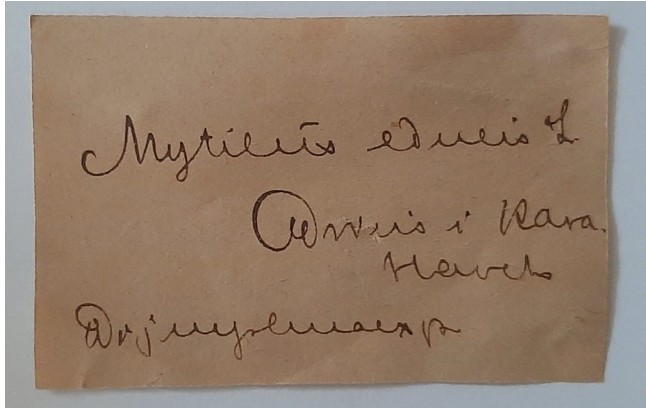

**FIG 1** Historical label for a museum specimen at the Zoological Museum of Copenhagen, Denmark. Museum documentation enabled curators to decipher and translate the Danish inscription into "*Mytilus edulis* L. Drift Ice in the Kara Sea. The Dijmphna Expedition." This documentation, however, leaves scarce contextual information useful for downstream analyses and interpretations, such as precise collection date and geographical coordinates, environmental conditions of the find, and postcollection processing. (Photo courtesy of Clio Der Sarkissian.)

generated from ancient metagenomic studies. However, it is something of which the community is aware (14, 23, 24) and needs to be carefully considered in the future as studies move in this direction. A major role in the control of ethical practices relating to aDNA studies can also be played by other actors, such as reviewers and panels considering grant applications and project officers of funding agencies, as well as journal editors and reviewers of scientific articles. Such individuals may check for proof of ethical clearance for all samples included in studies, e.g., a letter of intent with stakeholders, decisions from relevant ethical review boards (approval numbers), or authorizations/permits to analyze or export from museums or collections.

For aDNA researchers, several action-based recommendations have been made to improve ethical practices (22, 25). Access to samples should be conditioned on an agreement after discussions with stakeholders from the scientific (e.g., archaeologists, anthropologists, collection curators) and/or civil (e.g., local communities, indigenous peoples, descendants) community. Agreements can include aspects such as the perimeter of the project (e.g., objectives, team composition, host institution, methods with regard to the state of the art, and funding), the active participation of local researchers and community members, the conditions for sample storage or repatriation upon study completion, and the strategy to communicate progress and results both throughout the project and when releasing final articles and data sets. Existing documentation for such agreements should be provided upon submission of grant proposals or scientific articles. Ethical procedures should be shared within and outside our community so as to encourage widespread ethical practices and transparency in ancient metagenomics. Additionally, researchers working in ancient metagenomics should be encouraged, when offered, to take or develop ethics courses.

## ESSENTIAL METADATA AND WHERE TO FIND IT

Ancient DNA researchers are also confronted with the erosion of the information necessary to rigorously analyze and interpret ancient metagenomic data in their full temporal, biological, social, and cultural contexts. Metadata from long-dead individuals is often compiled indirectly from the curated documents at hand for past collections and excavations or from reconstructions based on previous and ongoing anthropological, archaeological, and historical analyses. These metadata can often be incomplete, inconsistent, difficult to access, and sometimes not trivially standardized (26), especially for samples opportunistically collected in the past (see Fig. 1 as an example). Therefore, reviewers should pay attention to the representation of such important but limited metadata in manuscripts and alert researchers who fail to provide information in an accessible manner, i.e., in machine-readable file formats following the fair, accessible, interoperable, and reproducible (FAIR) principles (27). While

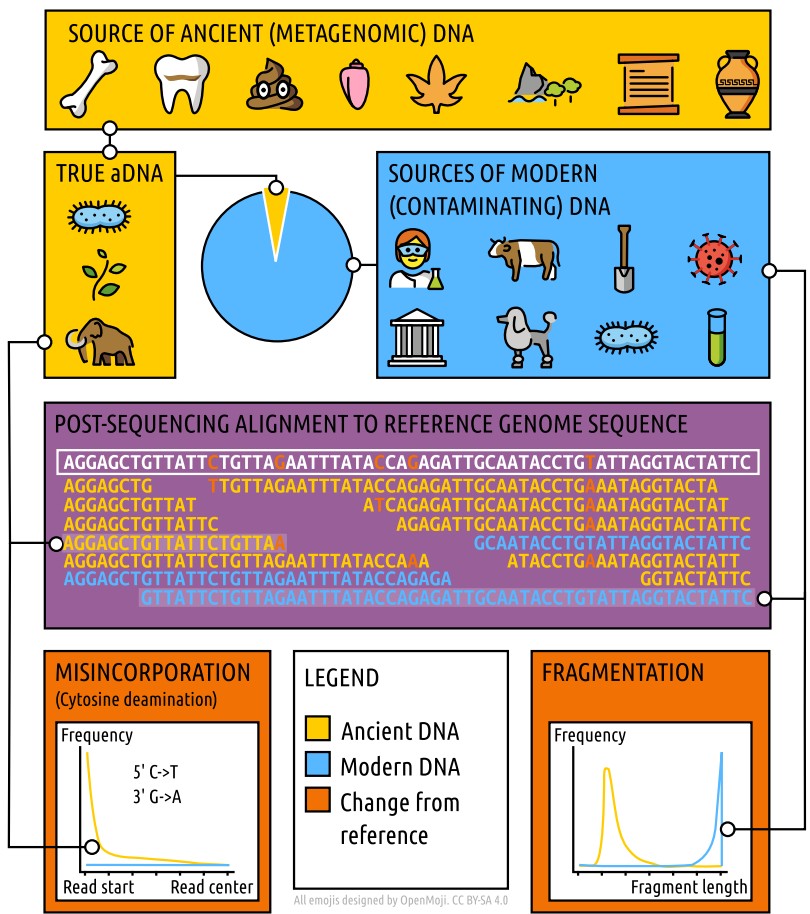

**FIG 2** Main characteristics and challenges of ancient metagenomic DNA. The original "endogenous" aDNA (i.e., the original DNA in the sample) of a wide range of sample types often makes up only a small fraction of the total genetic content of a sample. Contaminating environmental DNA from close relatives will often reduce confidence in taxonomic profiling and variant calling due to alignment of reads with variants from organisms other than the true ancient organism under investigation. True aDNA has a greater frequency of characteristic C-to-T deamination miscoding lesions at 5′ molecule termini (and a corresponding G-to-A mutation on the complementary 3′ strand of double-stranded libraries). True aDNA fragment distributions typically peak at short lengths, often around 30 to 70 bp, compared to the long lengths of modern DNA.

aDNA researchers have been celebrated for being at the forefront of raw data sharing (28), this is useful only if the corresponding metadata is made just as freely available.

To increase scientific yield from precious samples, it is important that ancient metagenomicists aim to maximize documentation prior to destruction (e.g., photography, photogrammetry, three-dimensional computed tomography [3D-CT] scanning, X-ray fluorescence [XRF] sediment scanning), to digitize and confirm information in historical museum records from other sources, or to combine complementary analyses (e.g., macrofossils, proteins, and stable isotopes, such as radiocarbon for dating) in their studies, project planning, and grant applications. The field should also come together to develop solutions to increase the linkability of these different types of data, allowing efficient incorporation into interdisciplinary analysis, but also standardizing reporting formats. Initiatives such as AncientMetagenomeDir, established by the SPAAM community (29), and developing context-specific "Minimum Information about any (x) Sequence" (MIxS) checklists (30), can act as templates for how this might be performed. By doing so, these initiatives will allow researchers to more efficiently compile relevant data for downstream analyses, as has been initiated in modern human microbiome research (31).

## COMPATIBILITY OF MOLECULAR LABORATORY METHODS

In a wet lab, the processing of ancient samples is challenging due to the degraded nature of aDNA molecules (Fig. 2). Standardization of optimized laboratory protocols

mSystems®

has allowed for the high throughput of many of the steps for successful sequencing of aDNA (3, 32–37). However, other methods that have gained popularity for modern sample types, such as Hi-C or long-read sequencing, require pure, high-quality, and long DNA molecules and therefore cannot be readily, if at all, applied to ancient samples.

Moreover, the features of aDNA mean that molecular tools, such as standardized DNA extraction kits that have been applied to most modern legacy data sets, e.g., the Human Microbiome Project (38), cannot be readily utilized for archaeological samples. While these commercial kits are optimized for the nuances of various modern sample types, even those optimized for low-biomass samples favor the retention of longer molecules and typically remove short fragments below 70 bp, making them inappropriate when targeting aDNA (Fig. 2; Table 1) (39). However, there is a growing number of studies comparing the suitabilities of different laboratory methods specifically for ancient metagenomics (Table 2).

Furthermore, while there are continuous improvements in molecular methods for limiting destruction and maximizing the output of molecular data from archaeological samples, these are often buried within large publications. As a field, we should therefore utilize resources such as protocols.io (https://www.protocols.io/; see, e.g., reference 40) or publish standalone method papers that would increase the visibility (and usability) of these improvements. Last, but not least, the low concentration of endogenous genetic material of our samples requires us to monitor lab contamination to confirm the validity of the results. Therefore, aside from having negative controls at each step, it is also more than necessary to publish these negative controls alongside the sequenced samples. This is in order to allow for a thorough evaluation of the results and improve reproducibility, as taxa present in the negative controls are sometimes removed from downstream sample data analyses.

## VALIDATION AND AUTHENTICATION OF ANCIENT DNA

A primary objective of any study of ancient metagenomics is to distinguish sequencing reads deriving from true aDNA molecules originally "endogenous" to the sample against a complex environmental background containing reads originating from both ancient and modern sources. We and others in the field (personal correspondence) have experienced reviews that outright dismiss ancient metagenomic studies as a novelty not to be taken seriously, despite the painstaking computational validation efforts included in the majority of studies.

As a first step, well-characterized degradation features of aDNA, i.e., damage patterns and high fragmentation, allow us to confidently identify true aDNA (Fig. 2) when it is combined with more general genomic reconstruction metrics, such as reasonable depth and breadth coverages. However, in cases of proposed exceptional preservation and in the absence of detectable or significant damage patterns, it is paramount that alternative lines of evidence for aDNA authenticity are combined. Other evidence can come from ancestral positioning and/or branch shortening of an ancient microbial genome in a phylogenetic tree, detection of established-species-specific genes or extrachromosomal genetic elements, radiocarbon dating of the material in question if the microbe is not associated with the environment (e.g., a host-restricted pathogen), or detection of microbial taxa reflecting the expected sample type (e.g., paleofeces contain predominantly gut taxa, instead of soil). Therefore, identifying sections in grants or publications that describe how aDNA will be, or has been, authenticated should be a critical part of any reviewer's checklist. Several reviews of validation techniques and examples of newly developed methods that may be useful resources for reviewers have been published, and some of these are summarized in Table 2.

While the above aDNA validation methods are well established in paleogenomics, the highly complex nature of ancient metagenomic data may require additional authentication steps. For example, postdepositional contaminating DNA from the environment can also display damage, as it is subjected to the same degradational processes as the DNA of interest (41). Take taxa such as *Clostridium botulinum*, whose toxin causes deadly illnesses in humans but is a widely existing member of the soil/depositional environment

and necrobiome (42–44). Therefore, insufficient understanding of the biological context of different taxa may result in falsely attributing the source of an organism of interest (45).

Finding solutions to such problems remains a primary goal for the field over the coming years. These might be, for example, ensuring that studies include environmental controls, such as associated skeletal material (for example bone for dental calculus) (see, e.g., references 46 to 48) or soil (see, e.g., reference 6), when available. In addition, for studies targeting specific microbial species, more routinely developing collaborative projects and publications that include experts on modern strains would be highly beneficial.

## CONSIDERATIONS FOR DATA ANALYSIS

Implementing modern bioinformatics methods requires an increasing number of assumptions to be met regarding the underlying data, such as the expected sequencing error distribution, which is often violated by aDNA data. Therefore, these improvements, as hugely beneficial as they might be for modern data analysis, often cannot be directly applied to aDNA.

One prominent example is the calling of single nucleotide variants (SNVs) from high-throughput sequencing data. In the last decade, there have been large improvements in correctly identifying SNVs by determining haplotypes by local reassembly (GATK HaplotypeCaller [49]) or by the exact evaluation of haplotypes rather than read alignments (FreeBayes [50]). Neither can be directly applied to aDNA data due to the DNA's increased frequency of C-to-T substitutions caused by postmortem cytosine deamination and its, on average, short read length, which interferes with the correct haplotype inference. This often forces us to stick to older methods, even deprecated ones, such as GATK UnifiedGenotyper. Therefore, it is important that reviewers consider whether a method is actually applicable to aDNA data prior to suggesting its usage. Additionally, it should be taken into account that DNA preservation, and thus the available amount of unique endogenous DNA molecules, might further limit the possibility of performing analyses, such as strain analysis, that typically require high sequencing depth.

However, these limitations should not be used as a scapegoat to justify scientific lagging in our field, as methods tailored for aDNA analysis do exist. For example, damage-aware variant callers (e.g., snpAD [51], AntCaller [52], and ARIADNA [53]) have been developed, primarily for human population genomics, but could also potentially be tested for metagenomic contexts. Hence, it is very important that we do spend the appropriate amount of time and energy on validating our analyses to show that they do not suffer from biases introduced by unfit methods and on developing tailored software when necessary (examples of existing benchmarking studies, software, and pipelines can be found in Table 2). To test the suitability of a method, simulations of ancient metagenomic sequencing data with programs such as gargammel (54) can provide the required framework for such validations. Reviewers should thus consider requesting such validation to ensure the high quality of the analysis, while we should prioritize these types of analyses, improving quality over quantity.

## CONCLUSIONS

The analysis of complex DNA content via shotgun high-throughput sequencing, also known as metagenomics, has recently been used on a wide range of source materials in numerous studies. Although very similar, analyses of complex ancient DNA content face some additional challenges which might not be immediately apparent and have led to a number of misunderstandings in recent peer review processes. The majority of these are related to either the samples included in the study, such as the preservation of their DNA and their limited availability, or the methods used in the laboratory or during analysis, which are often unusual from the point of view of a reviewer but are necessary when considering the characteristics of aDNA. Despite its particularities, analyzing aDNA samples provides a rich and valuable resource of complementary information that will also help the field of modern metagenomics to advance in the future.

We hope that this opinion/hypothesis provides a useful resource for future reviewers to be able to thoroughly evaluate ancient metagenomic studies, with respect to both their ethical conduct and the scientific quality of the presented research, while improving awareness of the challenges that go hand in hand with such samples. The list of topics discussed here is, by far, not exhaustive, and we hope that through the Standards, Precautions, and Advances in Ancient Metagenomics (SPAAM) community, we can further extend it in the future and will be able to establish standards in our field that ensure the reproducibility and the interoperability of future work. One of the strengths of the SPAAM community is the diversity of its members, and this is highlighted by the high degree of interdisciplinarity of their research, which spans from microbiology to evolutionary biology and from archaeology to history. We therefore cordially invite modern metagenomicists to participate in SPAAM (https://spaam-community.github.io/) to help us improve the communication between our closely related fields and increase the scientific yield of ancient metagenomic studies in the future.

## ACKNOWLEDGMENTS

We thank Jaelle C. Brealey, Becky Cribdon, Katerina Guschanski, Ophélie Lebrasseur, Aleksandar D. Kostic, Frank Maixner, Mohamed S. Sarhan, Laura S. Weyrich, Marsha Wibowo, and Sterling L. Wright for providing reviews and summaries of common review critiques. We also thank Peter D. Heintzman, Nikolay Oskolkov, Adam Koziol, Jessica Hider, Katerina Guschanski, Ophélie Lebrasseur, Anan Ibrahim, Miriam J. Bravo-Lopez, Eleanor J. Green, Siobhan Mor, Aida Andrades Valtueña, Christina Warinner, and members of the Department of Paleobiotechnology at the Leibniz Institute for Natural Product Research and Infections Biology Hans Knöll Institute for comments on the manuscript. Furthermore, we thank the attendees of SPAAM2 and the SPAAM community. We also thank Tom Gilbert, Ludovic Orlando, and Christina Warinner for further comments on the manuscript and their support for early career researchers.

We declare that we have no competing interests.

I.M.V. was supported by the Werner Siemens-Stiftung (Paleobiotechnology, awarded to Pierre Stallforth, Hans-Knöll Institute, and Christina Warinner, Max Planck Institute for Evolutionary Anthropology). A.K.F. was supported by the Danish National Research Foundation award (CEH-DNRF143 awarded to M. Thomas P. Gilbert, University of Copenhagen, Denmark). Å.J.V. is supported by Carlsbergfondet Semper Ardens grant CF18-1109 (to M. Thomas P. Gilbert, University of Copenhagen, Denmark). A.H. is funded by the Deutsche Forschungsgemeinschaft (DFG; German Research Foundation) under Germany's Excellence Strategy EXC 2051, project ID 390713860. J.A.F.Y. was supported by grant ERC-2015-StG 678901-FoodTransforms (to Philipp W. Stockhammer, Ludwig Maximilian University, Germany). J.A.F.Y., I.M.V., and A.H. are supported by the Max Planck Society. C.D.S. is supported by the Centre National de la Recherche Scientifique, France.

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
