## [Reviewer comments · mSystems]

Ancient Metagenomic studies - considerations for the wider scientific community

Clio Der Sarkissian, Irina Velsko, Anna Fotakis, Åshild Vågene, Alexander Hübner, and James Fellows Yates

Corresponding Author(s): James Fellows Yates, Max Planck Institute for Evolutionary Anthropology

Review Timeline:

Submission Date:	November 1, 2021
Editorial Decision:	November 2, 2021
Revision Received:	November 24, 2021
Accepted:	November 24, 2021

Editor: Jack Gilbert

Reviewer(s): The reviewers have opted to remain anonymous.

Transaction Report:

DOI: <https://doi.org/10.1128/msystems.01315-21>

November 2, 2021

Dr. James A. Fellows Yates
Max Planck Institute for Evolutionary Anthropology
Department of Archaeogenetics
Deutscher Pl. 6
Leipzig, Sachsen 04103
Germany

Re: mSystems01315-21 (Ancient Metagenomic studies - considerations for the wider scientific community)

Dear Dr. Fellows Yates:

Thank you for submitting your manuscript to mSystems. We have completed our review and I am pleased to inform you that, in principle, we expect to accept it for publication in mSystems. However, acceptance will not be final until you have adequately addressed the reviewer comments.

I really like this opinion piece and would like to see it published in mSystems. I have a few comments first that I would like to see addressed if at all possible.

First, in the first section - the language is very defensive, written as if responding to reviewer criticism, which obviously it is. I would like to see an example laid out of where due to the obvious constraints the data generated can not provide all the relevant controls and replications, and yet still the data are able to support the results and conclusions. Otherwise, as it stands it reads as if you are saying, 'look we know the work can never be perfect, but it's still valuable even if we cannot generate data that can be robustly interrogated' - which is fair, but I would like to see one or two examples where this is made clear.

There is only passing comments on the types of protocols that should be in place to ensure high data quality for these samples. IF this is dealt with elsewhere then it should be adequately references and called out in the text as a resource.

When dealing with ethical considerations it would be valuable to put some outline of how you would deal with profit sharing with the community from any IP generated. This is a major consideration especially when dealing with samples from people's ancestors or cultural artifacts.

"The continuous improvement of existing computational methods and the development of new ones boost the analysis of modern metagenomics data." This sentence is not necessary delete it. Start with "implementing modern bioinformatics methods require an increasingly number of assumptions.....etc."

The reference to Table 2 appears only in the section on validating aDNA - but not all the references in there are reflecting of validation. Are these the benchmarking ones? It is unclear without diving in deeper into each paper, what the Publication Type is referring to or whether it is useful for the non-expert.

Thank you for the privilege of reviewing your work. Below you will find instructions from the mSystems editorial office.

Preparing Revision Guidelines

Corresponding authors may join or renew ASM membership to obtain discounts on publication fees. Need to upgrade your

membership level? Please contact Customer Service at Service@asmusa.org.

Sincerely,

Jack Gilbert

Editor, mSystems

Journals Department
POINT-BY-POINT RESPONSES TO REVIEWS ON MANUSCRIPT
mSystems01315-21R1

MAX-PLANCK-GESellschaft

“Ancient Metagenomic studies - considerations for the wider scientific community”

Clio Der Sarkissian, Irina M. Velsko, Anna K. Fotakis, Åshild J. Vågene, Alexander Hübner and **James A. Fellows Yates***

* corresponding author

We received 5 main comments from the editor. Our responses are below, with the corresponding changes highlighted in **green** in the marked-up manuscript. We have made additional minor changes based on further feedback from the SPAAM community and ourselves (some overlapping with the reviewer), and these are highlighted in **yellow** in the marked-up manuscript.

RESPONSE TO THE REVIEW

Point 1: First, in the first section - the language is very defensive, written as if responding to reviewer criticism, which obviously it is. I would like to see an example laid out of where due to the obvious constraints the data generated can not provide all the relevant controls and replications, and yet still the data are able to support the results and conclusions. Otherwise, as it stands it reads as if you are saying, 'look we know the work can never be perfect, but it's still valuable even if we cannot generate data that can be robustly interrogated' - which is fair, but I would like to see one or two examples where this is made clear.

Response: This is a fair comment, and indeed detracts slightly from the main message of this piece. We have re-phrased several sentences (e.g. lines 56-57, 81, removal of a sentence at the beginning of line 96).

We have also added two examples of how results can be robust as aDNA research progresses despite initial low sample sizes (lines 101-105), or how such studies can still provide useful insights into other research questions even if the original objectives were not met (lines 113-117)

Point 2: There is only passing comments on the types of protocols that should be in place to ensure high data quality for these samples. IF this is dealt with elsewhere then it should be adequately references and called out in the text as a resource.

Response: Thank you for pointing this out. To emphasise that the consistent retrieval of high-quality aDNA is routinely possible, we have added a range of additional citations that correspond to well-established protocols for the retrieval of aDNA in general (line 195-196).

Additionally, we have extended Table 2 to include a range of laboratory-focused papers specifically for ancient metagenomics, such as assessing optimal methods for decontamination of modern DNA, or optimised enrichment methods for the recovery of ancient pathogen genomes. We now also refer to this table in the molecular laboratory methods paragraph (line 206-208).

Point 3: When dealing with ethical considerations it would be valuable to put some outline of how you would deal with profit sharing with the community from any IP generated. This is a major consideration especially when dealing with samples from people's ancestors or cultural artifacts.

Response: We thank you for this comment as this is currently a particularly pertinent topic in aDNA in general. This is a very complex and sensitive matter which we feel we cannot

fully address in this piece. There is indeed a lot of discourse around this topic mainly for human population genetic studies (e.g. most recently, Bardill et al. 2018 *Science*, Wagner et al. 2020 *Am J Hum Genet*, Alpaslan-Roodenberg et al. 2021 *Nature*), and it was extensively discussed at the SPAAM2 and SPAAM3 meetings.

Despite ongoing discussions, to our knowledge, very little has been published on this topic specifically for ancient metagenomics so far. In addition, we are currently unaware of good case studies on the use of ancient metagenomic data to generate ‘intellectual property’ (although this is certainly possible given the current expectation in the palaeogenomics field that all data is uploaded to public repositories).

We have therefore added a sentence explicitly stating this, citations to the meeting notes of the SPAAM2 and SPAAM3 workshops, and another reference (Curry 2021) that includes some discussion about the ethics involved in the study of ancient palaeofaeces. This includes a quote from a prominent researcher working on ethics in palaeogenomics, who states that, on the ethical front, ancient metagenomics is still in a ‘whole new grey area’ (lines 134-137).

Point 4: "The continuous improvement of existing computational methods and the development of new ones boost the analysis of modern metagenomics data." This sentence is not necessary delete it. Start with "implementing modern bioinformatics methods require an increasingly number of assumptions.....etc."

Response: We deleted the first line and rephrased the second, as requested (line 272).

Point 5: The reference to Table 2 appears only in the section on validating aDNA - but not all the references in there are reflecting of validation. Are these the benchmarking ones? It is unclear without diving in deeper into each paper, what the Publication Type is referring to or whether it is useful for the non-expert.

Response: We have now expanded the number of references in Table 2 to also cover laboratory protocols (see point 2).

We have inserted further references to Table 2 in the main text to correspond to each of the main publication types of Table 2 (reviews [line 63], wet-lab protocols [line 206-208], validation [original, line 254], benchmarking and software/pipelines [line 295-297])

ADDITIONAL CHANGES

The following changes have been made by the authors and based on feedback from additional members of the SPAAM community during the reviewing process.

Line 30-32: expanded the examples of issues that ancient metagenomic studies can encounter (less than optimal experimental design, confounding factors) to give further guidance as to contents of this manuscript.

Line 44-45: added example sample types to further what ancient metagenomic studies typically work on and provide further context to readers unfamiliar with ancient DNA.

Line 69-75: provided expanded description of the SPAAM2 meeting and the SPAAM community in general, to give more information regarding the motivation for this piece.

Line 89: moved reference to deeper sequencing being prohibitive to this paragraph from the following one as part of restructuring to make this paragraph less negative.

Line 91: added ‘if even possible’ to clarify that sometimes ancient metagenomics researchers work on ‘unique’ or ‘one of a kind’ samples where there is no further material.

Line 97: removed a sentence at the beginning of this paragraph due to redundancy with the previous paragraph.

Line 139, 141: extended list of people responsible for good ethical practises to also include collection/material curators (given no available explicit legal framework for ancient metagenomics samples).

Line 141: We have also changed the phrasing of requiring *completed* ethical approval from ethical review boards for grant applications, to just requesting ‘letter of intent’ from local stakeholder groups (e.g. indigenous communities), due to concerns by some in the SPAAM community that requiring full ethical permissions (which can be a long and complicated process) could penalise smaller labs that cannot begin full ethical application nor carry out research until they have funding in the first place.

Line 147: changed indigenous ‘groups’ to ‘peoples’ due to potentially problematic terminology.

Line 149: mention training into state-of-the-art methods so as to ensure communities can make fully informed decisions.

Lines 174-176: included further examples of analyses to also cover sediment and soils and other commonly used methods in archaeological science.

Line 182-183: citation of a recent initiative from modern human microbiome research as an example of how the ancient metagenomics community could develop such standardised metadata reporting initiatives.

Line 188: added that the museum label in Figure 1 is written in Danish, as this can sometimes be an Issue in English language-dominated academia.

Line 199: rephrased to ancient samples for clarity.

Line 217: add ‘more than’ to more strongly emphasize the importance of negative controls for ancient DNA due to contamination risks.

Line 218-220: expanded the description to explain why negative controls are so important for aDNA over other fields.

Line 229-230: (Figure 2 caption) specified some ancient samples can actually have high endogenous levels (e.g. dental calculus), and also clarified that different library building strategies will result in different damage patterns.

Line 244: typo correction (originally missing ‘patterns’).

Line 245: emphasised that *multiple* lines of authentication evidence *are* necessary.

Line 249: typo correction.

Line 259-261: emphasized that *C. botulinum* is a common soil contaminant despite being commonly known as a pathogenic organism. We add this further emphasis as it is commonly reported as a putative pathogen in ancient metagenomic taxonomic screening without further authentication.

MAX-PLANCK-GESellschaft

Line 266-268: we put more emphasis on bone samples being better environmental controls for samples such as dental calculus, because their elemental/organic makeup is more similar than sediment or soil, but also because they are more commonly available than soil as most museums or collection owners will not store soil-counterparts from excavation.

Line 269: deleted redundant word.

Line 300: deleted 'the'.

Line 314: deleted 'that'.

Line 317: rephrased to 'topics' as in this piece we do put forward a defined 'list' of action items.

Table 1: Fair Principles - typo correction to reusable. Authentication – updated to damage for consistency with the rest of the manuscript.

Table 2: General Review – typo correction. Computational tool – typo correction. Pipeline – typo correction.

List of abbreviations: added XRF Fluorescence in reference to line 171.

Funding: put in requested statement text from funder (Werner-Siemens Stiftung).

Acknowledgments: Added additional thanks to SPAAM community members and supervisors who gave additional feedback during the revision process.

References: updated based on additional citations added as requested.

November 24, 2021

Dr. James A. Fellows Yates
Max Planck Institute for Evolutionary Anthropology
Department of Archaeogenetics
Deutscher Pl. 6
Leipzig, Sachsen 04103
Germany

Re: mSystems01315-21R1 (Ancient Metagenomic studies - considerations for the wider scientific community)

Dear Dr. Fellows Yates:

Your manuscript has been accepted, and I am forwarding it to the ASM Journals Department for publication. For your reference, ASM Journals' address is given below. Before it can be scheduled for publication, your manuscript will be checked by the mSystems senior production editor, Ellie Ghatineh, to make sure that all elements meet the technical requirements for publication. She will contact you if anything needs to be revised before copyediting and production can begin. Otherwise, you will be notified when your proofs are ready to be viewed.

Publication Fees:

We recognize that the video files can become quite large, and so to avoid quality loss ASM suggests sending the video file via <https://www.wetransfer.com/>. When you have a final version of the video and the still ready to share, please send it to Ellie Ghatineh at eghatineh@asmusa.org.

Sincerely,

Jack Gilbert
Editor, mSystems

Journals Department
Phone: 1-202-942-9338